# Optimal duration of the apnea test for determining brain death: Benefit of the short-term apnea test

**Seung Min Baik**[1,2], **Jin Park**[3], **Tae Yoon Kim**[4], **Kyung Sook Hong**[4]*

1 Division of Critical Care Medicine, Department of Surgery, Ewha Womans University Mokdong Hospital, Ewha Womans University College of Medicine, Seoul, Korea, 2 Department of Surgery, College of Medicine, Graduate School of Korea University, Seoul, Korea, 3 Division of Critical Care Medicine, Department of Neurology, Ewha Womans University Seoul Hospital, Ewha Womans University College of Medicine, Seoul, Korea, 4 Division of Critical Care Medicine, Department of Surgery, Ewha Womans University Seoul Hospital, Ewha Womans University College of Medicine, Seoul, Korea

* hongks@ewha.ac.kr

**Data Availability Statement:** Data cannot be shared publicly for confidentiality reasons. Data are available from the Institutional Review Board (IRB) of Ewha Womans University Mokdong Hospital (contact the committee by email: irb-mok@eumc.

## Abstract

### Background

The criteria for brain death determination have not been unified globally, and there is no global consensus on the apnea test, which is essential for determining brain death. Since the apnea test is associated with many complications, we aimed to determine an optimal duration of the apnea test.

### Methods

We analyzed the results of the apnea test performed for brain death determination between August 2013 and February 2021 at a single institution in South Korea. Elevations in the partial pressure of carbon dioxide and mean arterial blood pressure fluctuations over time in the apnea test were recorded.

### Results

In the 1st and 2nd tests, the mean partial pressure of carbon dioxide increased by more than 20 mmHg at 3 min after the apnea test compared to before the test ($P < 0.05$). At 4 min in the 1st test and 5 min in the 2nd test, the partial pressure of carbon dioxide exceeded 60 mmHg ($P < 0.05$). The fluctuation in the mean arterial blood pressure observed for 5 min during the apnea test was not significant. There was no significant fluctuation in the mean arterial blood pressure over time in the apnea test between patients with normal chest radiography findings and those with abnormal chest radiography findings ($P = 0.888$).

### Conclusions

Our study proposes that a short-term apnea test protocol is valid for the preservation of organs for donation.

ac.kr) for researchers who meet the criteria for accessing confidential data.

**Funding:** The author(s) received no specific funding for this work.

**Competing interests:** The authors have declared that no competing interests exist.

# Introduction

Brain death was first described in 1959 and is defined as the irreversible loss of all identifiable brain functions, including those of the brainstem [1,2]. Although it is an established medical concept, there is no global consensus regarding the determination of brain death [3]. According to the American Academy of Neurology (AAN) guidelines, the following criteria must be met when determining brain death: presence of an irreversible etiology; neurologically confirmed coma, loss of the brainstem reflex, and apnea test positivity; and positive optional tests (no cerebral blood flow on angiography or transcranial Doppler ultrasound [TCD], no electrical activity on electroencephalography [EEG], and no uptake of technetium on brain scan) [4].

The apnea test is essential for clinically determining brain death [4]. Its main purpose is to demonstrate the absence of a respiration control system reflex in the brainstem when stimulation of respiration occurs with an increased partial pressure of carbon dioxide ($PaCO_2$). Regarding the apnea test and positive criteria for the apnea test in South Korea [5], pre-oxygenation is first achieved with 100% oxygen ($O_2$) or 95% $O_2$ and 5% $CO_2$ via a mechanical ventilator for 10 min. Thereafter, the mechanical ventilator is removed, and 6 L/min of 100% $O_2$ is supplied through an endotracheal tube. If spontaneous respiration is not induced although $PaCO_2$ has risen to ≥50 mmHg as determined by blood gas analysis, the apnea test is deemed positive. The apnea test in the United States slightly differs from that in South Korea [4]. In the United States, after removing the mechanical ventilator, there is observation for an apnea pattern for 8 to 10 minutes. Thereafter, arterial blood gas analysis (ABGA) is performed, and the apnea test is considered positive if $PaCO_2$ exceeds 60 mmHg or increases by ≥20 mmHg above the reference value. The differences between the apnea tests of the two countries mainly focus on the test duration as well as $PaCO_2$, which is a criterion for determining a positive apnea test. In Japan, the standard duration for the apnea test is unspecified, but the $PaCO_2$ is the same as in the United States [6].

A survey of 80 countries showed no agreement between continents and countries, and even within one country regarding brain death diagnostic criteria and apnea test modalities for brain death [3]. There were differences regarding existing laws and guidelines related to determining brain death, number of medical staff for the determination, observation duration, test for determination, and apnea test procedure. Factors other than the implementation patterns of the apnea test are closely related to ethical considerations. In other words, there is no global consensus on the implementation of the apnea test, although such a consensus seems necessary for patient safety. The implementation pattern of the apnea test is more closely related to the condition of the patient waiting for brain death determination. This may be due to complications of the apnea test, which include hypoxemia, hypotension, acidemia, hypercapnia, increased intracranial pressure, pulmonary hypertension, and arrhythmias [7–14]. Compared to the conventional apnea test, the modified apnea test (MAT) maintains positive end-expiratory pressure (PEEP) and can prevent lung atelectrauma and hypoxia even after mechanical ventilator removal [10]. However, MAT does not reduce other hypercapnia- and hypoxia-induced complications. These complications can result in damage to organs intended for donation. Since determining brain death is related to stopping unnecessary life support and also to organ donation, organ preservation should be considered while managing potential brain death patients for organ donation [15]. Therefore, identifying the most effective and safest apnea test method that allows the maintenance of hemodynamic stability without interfering with the brain death determination process can result in better post-transplantation outcomes.

The study aimed to suggest an appropriate apnea test duration by reviewing the records of a single institution in South Korea.

## Materials and methods

### Patients and collected data

Patients who underwent brain death management for organ donation between August 2013 and February 2021 were enrolled. The most appropriate routine protocol for brain death management was implemented for all patients. The following demographic and clinical data were obtained: sex, age, cause of brain death, total duration of hospitalization, brain death management period, Acute Physiologic Assessment and Chronic Health Evaluation (APACHE) II score, plateau pressure, norepinephrine infusion rate, and chest radiography findings. Chest radiography findings were classified into normal and abnormal findings. The abnormal findings group included one or more of the following: pneumonia, pulmonary edema, pleural effusion, atelectasis, bronchiectasis, and emphysema. In addition, when cardiac donation was planned or cardiac function evaluation was necessary during management, echocardiography was performed, and ejection fraction (EF) values were recorded.

### Apnea test protocol

The apnea test protocol used in our institution employs MAT. The apnea test to determine brain death was performed twice, at least 6 hours apart. Ten minutes before the apnea test, pre-oxygenation was performed with the fraction of inspired oxygen ($FiO_2$) set to 100%. In the presence of the attending physician, the mechanical ventilator was removed from the patient, and a bag valve mask equipped with a PEEP valve was connected. The $O_2$ supplied had an $FiO_2$ of 100% and flow rate of 6 L/min. After applying the bag valve mask to the patient, we confirmed that the patient had apnea and performed an ABGA test every minute from 1 min to 5 min or more if possible. The attending physician monitored the patient's condition during the apnea test and recorded the arterial blood pressure (ABP) and pulse rate every minute during blood sampling. When the $PaCO_2$ level met the positive criteria for an apnea test, blood sampling was stopped and a mechanical ventilator was connected.

### Statistical analysis

All numeric variables, such as age, ABGA results, and variables related to hemodynamic status, are expressed as mean ± standard deviation. Categorical variables, such as sex, cause of brain death, and chest radiography findings, were analyzed using descriptive statistics. ABGA results, serum lactate level, and hemodynamic status observed every minute, were analyzed and compared with baseline results using the paired t-test. Serial changes in $PaCO_2$ and partial pressure of oxygen ($PaO_2$) in the 1st and 2nd apnea tests were analyzed by one-way repeated measures analysis of variance (ANOVA). Additionally, differences in plateau pressure and norepinephrine infusion rate between the pre- and post-apnea tests were analyzed using a paired t-test. The serial effect of differences in chest radiography findings on hemodynamic status during the apnea test was analyzed by two-way repeated-measures ANOVA. The statistical analysis was conducted and graphs were created using SPSS version 26.0 (IBM, Armonk, NY, USA). Statistical significance was designated as a significance level ($P$ value) less than 0.05.

### Ethics

This study was approved by the Institutional Review Board (IRB) of Ewha Womans University Mokdong Hospital (approval number: EUMC 2021-02-027). The requirement of informed consent was waived because of the retrospective design of the study.

## Results

Eighty-six patients who underwent the apnea test to determine brain death between August 2013 and February 2021 were enrolled in this study. Among them, 55 (64%) were men and 31 (36%) were women. The average age of the patients was 51.8 ± 13.8 years. Non-traumatic hemorrhage was the most common cause of brain death (n = 33, 38.4%). The total duration of hospitalization and the brain death management period were 14.4 ± 32.8 days and 2.8 ± 0.9 days, respectively. A mechanical ventilator was applied to all patients, and the average plateau pressure was 20.40 ± 5.33 cmH$_2$O. On chest radiography, 21 patients (24.4%) had normal findings and 65 patients (75.6%) had abnormal findings. Echocardiography was performed in 73 of the 86 enrolled patients: 16 (21.9%) and 57 (78.1%) patients had EF <50% and ≥50%, respectively. The demographic and clinical characteristics of the patients are presented in Table 1. In the 1$^{st}$ apnea test, 3 patients underwent the test for up to 6 min, 2 patients for up to 8 min, 1 patient for up to 9 min, and 1 patient for up to 10 min. In the 2$^{nd}$ apnea test, 5 patients underwent the test for up to 6 min and 1 patient for up to 7 min.

There were significant changes in pH and PaCO$_2$ levels compared to the baseline results. At 3 minutes in the 1$^{st}$ and 2$^{nd}$ apnea tests, PaCO$_2$ exceeded the baseline value by 20 mmHg ($P < 0.05$) (Table 2). There was no significant change in the mean ABP during the 5-minute apnea test. The ABGA results, hemodynamic status, and serum lactate levels observed during the apnea test are presented in Table 2. In this study, the PaCO$_2$ level exceeded 60 mmHg within 5 min in 76 (88.4%) and 81 (94.2%) cases of the 1$^{st}$ and 2$^{nd}$ apnea tests, respectively.

**Table 1. Demographics and clinical characteristics.**

| Variables | n = 86 |
|---|---|
| Sex (male:female) | 55 (64%):31 (36%) |
| Age (yr) | 51.8±13.8 |
| BMI (kg/m$^2$) | 23.22±3.67 |
| Cause of brain death | |
| Traumatic intracranial hemorrhage | 18 (20.9%) |
| Non-traumatic intracranial hemorrhage | 33 (38.4%) |
| Cerebral infarction | 3 (3.5%) |
| Encephalitis | 2 (2.3%) |
| Cardiac arrest | 12 (14.0%) |
| Hanging-induced hypoxic brain injury | 17 (19.8%) |
| Drawning-induced hypoxic brain injury | 1 (1.2%) |
| Total length of hospitalization (days) | 14.4 ±32.8 |
| Brain death management period (days) | 2.8±0.9 |
| APACHE II score | 31.6±7.1 |
| Plateau pressure (cmH$_2$O) | 20.40±5.334 |
| Norepinephrine infusion rate (mcg/kg/min) | 0.14±0.13 |
| Chest X-ray finding | |
| Normal finding | 21 (24.4%) |
| Abnormal findings[a] | 65 (75.6%) |
| Ejection fraction on echocardiography (n = 73) | |
| <50% | 16 (21.9%) |
| ≥50% | 57 (78.1%) |

BMI: Body mass index; APACHE II: Acute Physiologic Assessment and Chronic Health Evaluation.

[a]Abnormal findings: Pneumonia, pulmonary edema, pleural effusion, atelectasis, bronchiectasis and emphysema.

**Table 2. 1st and 2nd apnea test results.**

| Variables | Values[a] | ΔValues[a] | | | | |
|---|---|---|---|---|---|---|
| ABGA | Baseline results | Δ1minute results (P Value) | Δ2minute results (P Value) | Δ3minute results (P Value) | Δ4minute results (P Value) | Δ5minute results (P Value) |
| **1st apnea test** | | | | | | |
| pH | $7.323 \pm 0.081$ | $-0.066 \pm 0.040$ ($< 0.05^*$) | $-0.106 \pm 0.023$ ($< 0.05^*$) | $-0.138 \pm 0.034$ ($< 0.05^*$) | $-0.152 \pm 0.031$ ($< 0.05^*$) | $-0.166 \pm 0.050$ ($< 0.05^*$) |
| $PaCO_2$ (mmHg) | $41.4 \pm 4.2$ | $13.01 \pm 4.27$ ($< 0.05^*$) | $15.66 \pm 4.36$ ($< 0.05^*$) | $20.60 \pm 5.81$ ($< 0.05^*$) | $22.59 \pm 6.62$ ($< 0.05^*$) | $25.18 \pm 9.07$ ($< 0.05^*$) |
| $PaO_2$ (mmHg) | $327.0 \pm 150.3$ | $56.05 \pm 86.74$ ($< 0.05^*$) | $-47.17 \pm 103.22$ ($< 0.05^*$) | $-59.77 \pm 159.35$ ($< 0.05^*$) | $-114.32 \pm 113.05$ ($< 0.05^*$) | $-88.89 \pm 98.75$ ($< 0.05^*$) |
| $HCO_3$ (mEq/L) | $21.2 \pm 3.8$ | $2.59 \pm 1.91$ ($< 0.05^*$) | $2.03 \pm 2.18$ ($< 0.05^*$) | $1.91 \pm 1.60$ ($< 0.05^*$) | $1.41 \pm 5.41$ (0.137) | $2.15 \pm 1.89$ ($< 0.05^*$) |
| $SaO_2$ (%) | $97.9 \pm 3.5$ | $0.89 \pm 2.71$ (0.168) | $-2.49 \pm 9.74$ (0.115) | $-3.78 \pm 7.54$ ($< 0.05^*$) | $-2.42 \pm 5.64$ ($< 0.05^*$) | $-3.41 \pm 9.18$ (0.187) |
| Mean arterial blood pressure (mmHg) | $97 \pm 21$ | $9.24 \pm 26.12$ (0.164) | $4.91 \pm 17.48$ (0.106) | $-1.61 \pm 22.82$ (0.635) | $-10.657 \pm 34.70$ (0.074) | $-6.62 \pm 30.92$ (0.455) |
| Pulse rate (bpm) | $99 \pm 25$ | $3.00 \pm 6.36$ (0.070) | $2.34 \pm 10.79$ (0.208) | $3.80 \pm 10.02$ ($< 0.05^*$) | $4.09 \pm 9.85$ ($< 0.05^*$) | $0.77 \pm 11.02$ (0.806) |
| Serum lactate level (mg/dL) | $13.9 \pm 7.9$ | $0.19 \pm 2.21$ (0.713) | $-0.59 \pm 1.78$ ($< 0.05^*$) | $-1.12 \pm 2.25$ ($< 0.05^*$) | $-0.90 \pm 2.20$ ($< 0.05^*$) | $-1.60 \pm 3.28$ (0.157) |
| **2nd apnea test** | | | | | | |
| pH | $7.377 \pm 0.065$ | $-0.081 \pm 0.026$ ($<0.05^*$) | $-0.111 \pm 0.028$ ($< 0.05^*$) | $-0.134 \pm 0.034$ ($< 0.05^*$) | $-0.149 \pm 0.052$ ($< 0.05^*$) | $-0.162 \pm 0.040$ ($<0.05^*$) |
| $PaCO_2$ (mmHg) | $39.7 \pm 3.1$ | $11.58 \pm 4.44$ ($< 0.05^*$) | $16.76 \pm 6.38$ ($< 0.05^*$) | $30.29 \pm 69.62$ ($< 0.05^*$) | $37.72 \pm 92.21$ ($< 0.05^*$) | $23.03 \pm 7.87$ ($< 0.05^*$) |
| $PaO_2$ (mmHg) | $379.2 \pm 186.1$ | $30.62 \pm 120.06$ (0.324) | $-44.32 \pm 111.41$ ($< 0.05^*$) | $16.03 \pm 462.97$ (0.817) | $-123.06 \pm 124.21$ ($< 0.05^*$) | $-118.56 \pm 208.01$ (0.088) |
| $HCO_3$ (mEq/L) | $23.1 \pm 4.0$ | $1.39 \pm 1.71$ ($< 0.05^*$) | $2.08 \pm 2.22$ ($< 0.05^*$) | $6.71 \pm 31.08$ (0.150) | $2.18 \pm 1.57$ ($< 0.05^*$) | $2.03 \pm 1.90$ ($< 0.05^*$) |
| $SaO_2$ (%) | $98.7 \pm 1.9$ | $-0.28 \pm 2.41$ (0.624) | $-1.04 \pm 4.76$ (0.157) | $-2.34 \pm 7.10$ ($< 0.05^*$) | $-1.26 \pm 2.51$ ($< 0.05^*$) | $-1.81 \pm 4.72$ (0.211) |
| Mean arterial blood pressure (mmHg) | $96 \pm 17$ | $6.87 \pm 27.22$ (0.345) | $4.55 \pm 23.37$ (0.232) | $0.00 \pm 26.45$ (1.000) | $-0.35 \pm 26.11$ (0.938) | $-1.73 \pm 26.68$ (0.842) |
| Pulse rate (bpm) | $96 \pm 17$ | $-0.33 \pm 8.34$ (0.879) | $1.00 \pm 6.25$ (0.324) | $2.70 \pm 8.78$ (0.050) | $2.77 \pm 6.71$ ($< 0.05^*$) | $5.56 \pm 6.84$ ($< 0.05^*$) |
| Serum lactate level (mg/dL) | $13.3 \pm 8.6$ | $-0.80 \pm 1.02$ ($< 0.05^*$) | $-0.76 \pm 1.62$ ($< 0.05^*$) | $-0.77 \pm 1.82$ ($<0.05^*$) | $-1.13 \pm 1.43$ ($< 0.05^*$) | $-0.83 \pm 2.07$ (0.262) |

$PaCO_2$: Partial pressure of carbon dioxide; $PaO_2$: Partial pressure of oxygen; $HCO_3$: Bicarbonate; $SaO_2$: Oxygen saturation.

$^*P < 0.05$.

[a]Values expressed as mean ± SD.

The serial analysis of $PaCO_2$ for 5 min showed that the increase in $PaCO_2$ was significant over time in the 1st and 2nd apnea tests. $PaCO_2$ exceeded 60 mmHg in 3 min in the 1st apnea test and 4 min in the 2nd apnea test ($P < 0.05$) (Fig 1).

The serial analysis of $PaO_2$ for 5 min showed that in both the 1st and 2nd apnea tests, $PaO_2$ was elevated at 1 min of the test, but decreased thereafter. The change trend was significant in the 1st apnea test ($P < 0.05$), but not in the 2nd apnea test ($P = 0.095$) (Fig 2).

The norepinephrine infusion rate was significantly increased between the 1st pre- and post-apnea tests ($0.13 \pm 0.13$ μg/kg/min in the pre-apnea test vs $0.15 \pm 0.13$ μg/kg/min in the post-apnea test, $P < 0.05$). Except for this result, there was no significant difference in plateau pressure and norepinephrine infusion rate in the pre- and post-apnea tests (Table 3).

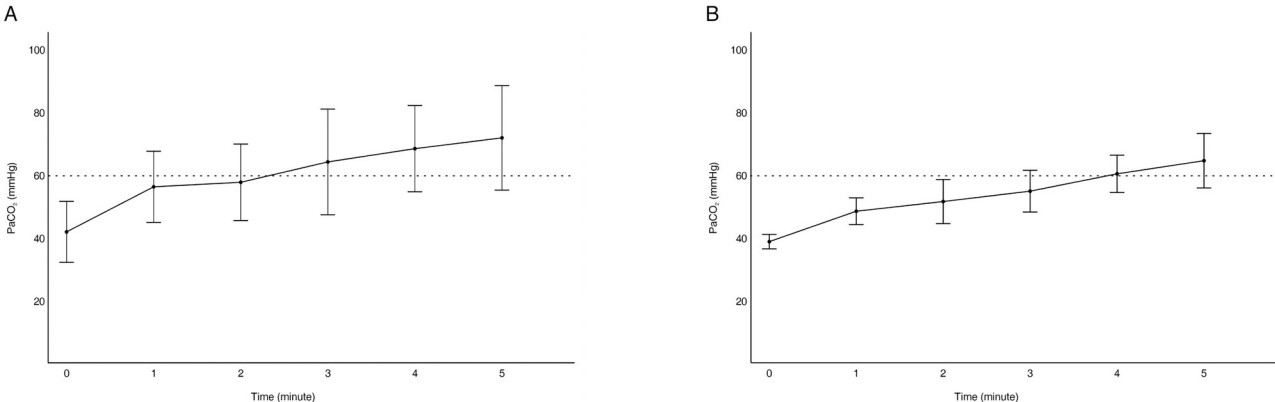

**Fig 1. Increase in the partial pressure of carbon dioxide ($PaCO_2$) in the apnea test.** (A) At 3 min in the 1st apnea test, the $PaCO_2$ level was above 60 mmHg. (B) At 4 min in the 2nd apnea test, the $PaCO_2$ level was above 60 mmHg. The apnea test positive criterion is $PaCO_2 > 50$ mmHg in South Korea, and $PaCO_2 > 60$ mmHg according to the American Academy of Neurology (AAN) guidelines. The dotted line denotes the apnea test positive criterion according to the AAN guidelines.

Subgroup analysis was also performed. Changes in mean ABP during the apnea test were compared between the normal and abnormal finding groups on chest radiography, and there was no significant difference between the two groups ($P = 0.888$) (Fig 3).

## Discussion

This study analyzed the results of apnea tests conducted by a single institution in South Korea and suggests a rational and unified international guide to the apnea test protocol.

According to the results, $PaCO_2$ exceeded 60 mmHg in ABGA within 4 min of starting the apnea test (Figs 1 and 2). This result met the positive criteria for the apnea test suggested by the AAN guidelines. In this study, the increase in $PaCO_2$ over time was significant within 5 min. The short-term apnea test is considered an essential test for brain death determination.

Elevation of $PaCO_2$ causes an additional increase in intracranial pressure in patients with potential brain death, as well as complications such as weakening of myocardial contractility, arrhythmias, and respiratory acidosis [16]. According to a review of apnea test complications

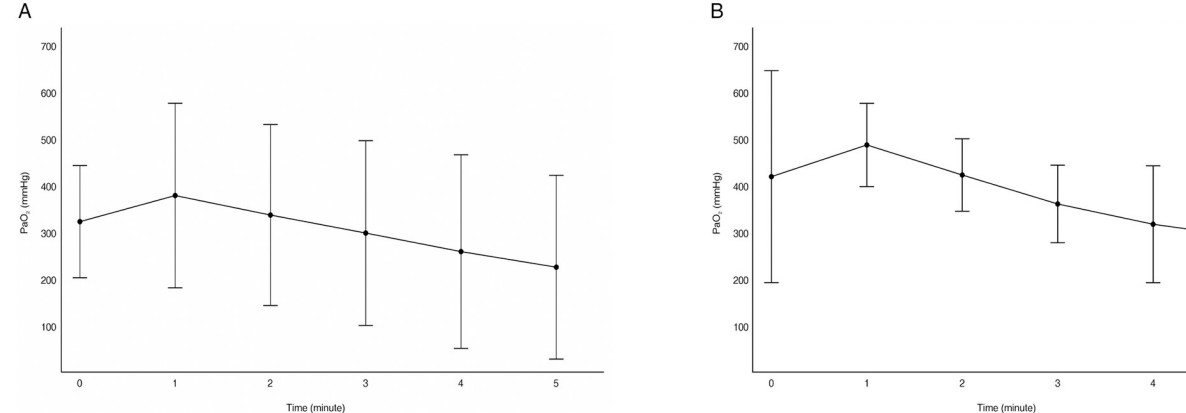

**Fig 2. Serial change in partial pressure of oxygen ($PaO_2$).** (A) Serial changes in $PaO_2$ were significant in the 1st apnea test ($P < 0.05$). (B) Serial changes in $PaO_2$ were not significant in the 2nd apnea test ($P = 0.095$).

**Table 3. Plateau pressure and norepinephrine infusion rate of pre- and post-apnea test.**

| Variables | Values | P Value |
| --- | --- | --- |
| 1st apnea test | | |
| Plateau pressure (cmH$_2$O) | | |
| Pre-apnea test | 20.66±5.24 | 0.072 |
| Post-apnea test | 22.72±6.16 | |
| Norepinephrine infusion rate (mcg/kg/min) | | |
| Pre-apnea test | 0.13±0.13 | < 0.05* |
| Post-apnea test | 0.15±0.13 | |
| 2nd apnea test | | |
| Plateau pressure (cmH$_2$O) | | |
| Pre-apnea test | 22.76±6.17 | 0.397 |
| Post-apnea test | 22.42±5.95 | |
| Norepinephrine infusion rate (mcg/kg/min) | | |
| Pre-apnea test | 0.09±0.10 | 0.120 |
| Post-apnea tes | 0.09±0.11 | |

*$P < 0.05$.

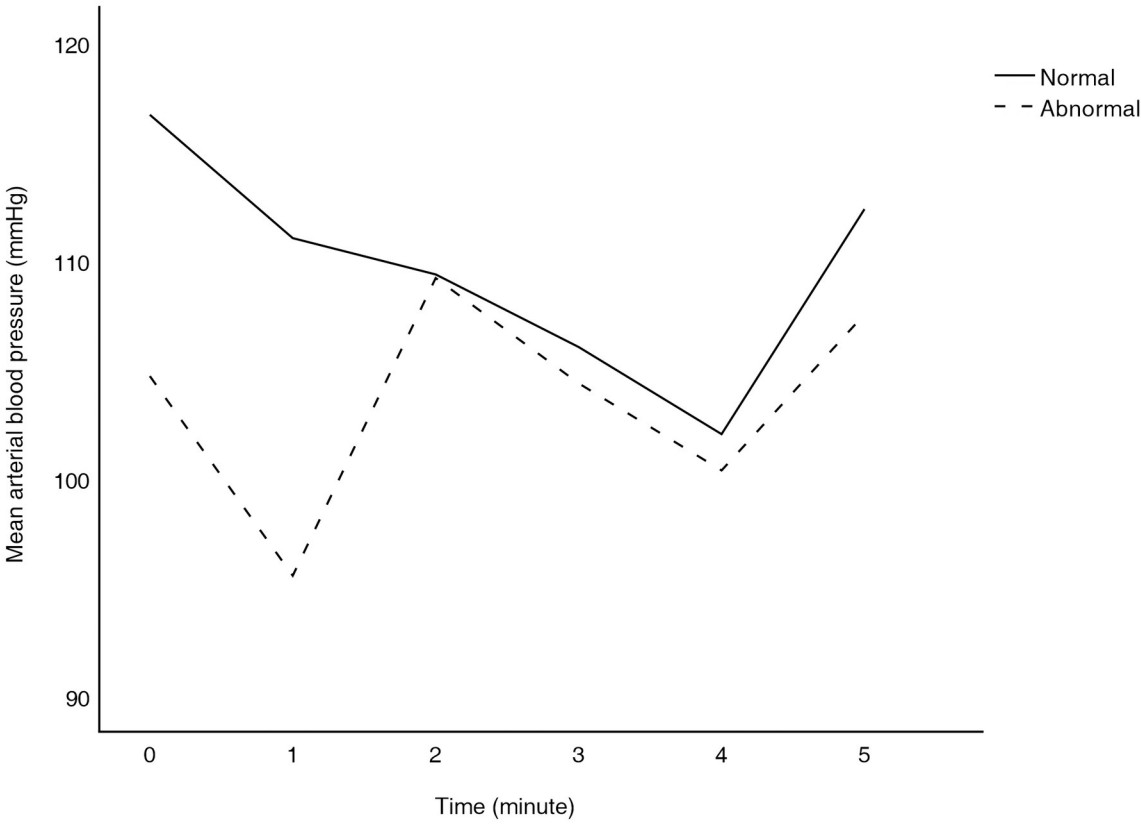

**Fig 3. Difference in mean arterial blood pressure fluctuation during the apnea test according to chest radiography findings.** The difference in mean arterial blood pressure fluctuations between the two groups was not significant ($P = 0.888$).

reported in 2013, hypotension was observed in 111 (18%) of 608 patients who underwent the test [17]. However, maintaining proper blood pressure in the management of brain death is important to prevent ischemic changes in organ(s) for donation and to increase transplant success rates. Therefore, according to the AAN guidelines, the systolic blood pressure should be $\geq$ 100 mmHg and the mean ABP should be $\geq$ 60 mmHg before starting the apnea test for brain death determination. In this study, we aimed to maintain a mean ABP of $\geq$ 65 mmHg before the apnea test and during the entire management period. In fact, the pH from the ABGA during or immediately after the apnea test decreased significantly over time. There were no significant differences in the mean ABP and pulse rate over time. Cardiovascular dysfunction, which is a common complication during the short-term apnea test and lasted for 1 to 5 minutes, was not severe. Therefore, a short apnea test should be considered for safer outcomes.

The vasopressor infusion rate was significantly increased between the 1[st] pre- and post-apnea tests. However, the increase in vasopressor infusion rate in the 1[st] apnea test was as small as 0.02 μg/kg/min. According to the brain death management protocol of our institution, we checked the mean ABP before the apnea test and, in some cases, preemptively increased the vasopressor injection rate when a borderline mean ABP of approximately 65 mmHg was observed. Therefore, it might have acted as a bias due to the external factors of the study. In addition, it is possible that these results were observed because fluid resuscitation was not sufficiently administered to manage potential brain death patients before the 1[st] apnea test for brain death determination. In fact, the increase in vasopressor infusion rate was not significant between the 2[nd] pre- and post-apnea tests.

During the apnea test, there was no significant difference in mean ABP fluctuations between the normal and abnormal finding groups on chest radiography; therefore, a short-term apnea test seems safe even in patients with poor lung conditions.

MAT is a method for preventing lung damage without affecting hypercapnia by maintaining PEEP even after removing the mechanical ventilator from the patient [11,18]. In the 1[st] apnea test in this study, PaO$_2$ was significantly changed every minute, but its value was $\geq$ 90 mmHg (Table 2). Previous studies have confirmed that MAT did not negatively affect apnea test results and allowed a safer completion of the tests [19,20]. MAT was also performed in the present study. Changes in plateau pressure in the pre- and post-apnea tests were checked as a parameter to confirm the presence or absence of lung damage, and no significant changes were observed. Changes in plateau pressure disproved the changes in lung compliance. Although the degree of change was not significant in this study, if the duration of the apnea test is prolonged, lung compliance may worsen. If lung donation is planned, caution is required during the apnea test.

If a normal body temperature is maintained and there is no lung disease, only 5 min of apnea can raise PaCO$_2$ from 40 mmHg to 60 mmHg [21]. Nevertheless, a global consensus has not yet been reached. Although the apnea test in South Korea can be considered to be a relatively weak standard compared to the AAN guideline, EEG is mandatory for all brain death determinations, and TCD is also performed in some cases. However, according to the AAN guidelines and a publication by the World Brain Death Project in 2020, EEG was excluded from the mandatory tests because of its high false-positive rate; however, it could be performed as an optional test if the apnea test is unavailable [4,22]. TCD, which is not an essential test for brain death determination, can also be performed as an optional test if the apnea test is infeasible. The sensitivity and specificity of TCD for determining brain death are 90% and 98%, respectively [23]. The additional use of TCD may help shorten the duration of the apnea test in brain death determination.

In a previous study, we attempted to determine the optimal duration of the apnea test by performing ABGA every minute during the apnea test [10]. In that study, $PaCO_2$ exceeded 60 mmHg 4 min after the removal of the mechanical ventilator; thus, it can be considered that observing the patient's apnea pattern for at least 8 min during the apnea test, as presented by the AAN guidelines, is relatively long. Although many studies have reported that complications such as hemodynamic instability and lung damage can be caused by the apnea test, studies on the possible shortening the apnea test duration are insufficient. The reason may be because of the recognition that more stringent criteria should be applied because the apnea test is valuable as an essential test for brain death determination.

A study of guidelines for brain death in 80 countries found that detailed guidelines were inconsistent, and in the U.S., there was no internal agreement because of differences in laws even between states [3]. Likewise, the implementation patterns of the apnea tests and positivity criteria have not been standardized worldwide. It will be difficult to achieve a consensus regarding the criteria for brain death determination because of differences in legal systems, culture, and medical standards between countries. Nevertheless, for the safety of patients waiting for brain death determination and for organ preservation after brain death determination, there is a need for a global consensus on the implementation pattern of the apnea test, which is generally considered essential in brain death determination. We suggest an optimal apnea test protocol as follows: 1) after removing the mechanical ventilator from the patient, observe the patient's apnea patterns for 5 min; 2) after 5 min, perform the short-term ABGA for 1 or 2 min; and 3) when it is confirmed that $PaCO_2$ exceeds 60 mmHg, terminate the apnea test immediately.

Because this study was conducted in a single institution, the apnea test was not performed on patients of various races, physiques, etc. This is an important limitation with regarding to generalizing our findings. A global multicenter study is necessary to establish a reasonable apnea test that can be widely performed worldwide.

## Conclusions

When brain death was determined, MAT for approximately 5 min resulted in a sufficient increase in $PaCO_2$, meeting the positive criteria for the apnea test, and hemodynamic instability during the test was not significant. This study should be valuable for providing guidelines for a globally relevant optimal duration for the apnea test.

## Acknowledgments

We would like to thank Professor Young-Joo Lee for her lecture on the management of brain death patients. We would also like to thank the Korea Organ Donation Agency staff for their assistance in the management of brain death patients.

## Author Contributions

**Conceptualization:** Seung Min Baik, Kyung Sook Hong.

**Data curation:** Seung Min Baik, Jin Park, Tae Yoon Kim, Kyung Sook Hong.

**Formal analysis:** Seung Min Baik, Kyung Sook Hong.

**Investigation:** Seung Min Baik.

**Methodology:** Seung Min Baik, Jin Park, Tae Yoon Kim, Kyung Sook Hong.

**Supervision:** Kyung Sook Hong.

**Visualization:** Seung Min Baik.

**Writing – original draft:** Seung Min Baik.

**Writing – review & editing:** Seung Min Baik, Jin Park, Tae Yoon Kim, Kyung Sook Hong.

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
