## [Decision Letter · Decision Letter 0]

14 Jun 2022

PONE-D-22-01880Optimal Duration of the Apnea Test for Determining Brain Death: Benefit of the Short-Term Apnea TestPLOS ONE

Dear Dr. Hong,

Thank you for submitting your manuscript to PLOS ONE. After careful consideration, we feel that it has merit but does not fully meet PLOS ONE’s publication criteria as it currently stands. Therefore, we invite you to submit a revised version of the manuscript that addresses the points raised during the review process.

Editor - Thank you for submitting your paper to us for review.  I sent it to three distinguished referees for comment and decision, and then three more of whom two agreed to review; you will see these below.  They thought that the paper has merit, but each have raised some issues to be addressed in a revision.  Please carefully consider the comments below and reply directly to each in a cover letter with appropriate marked and linked changes to the manuscript.  I look forward to seeing the revision, which I will handle personally for timeliness.  Please understand that this is not a guarantee of future publication, as the revised manuscript itself must stand on its own merit.

We look forward to receiving your revised manuscript.

Kind regards,

Steven Eric Wolf, MD

Academic Editor

PLOS ONE

Journal Requirements:

a) Did participants provide their written or verbal informed consent to participate in this study?

3. We note that you have stated that you will provide repository information for your data at acceptance. Should your manuscript be accepted for publication, we will hold it until you provide the relevant accession numbers or DOIs necessary to access your data. If you wish to make changes to your Data Availability statement, please describe these changes in your cover letter and we will update your Data Availability statement to reflect the information you provide

Reviewers' comments:

Reviewer's Responses to Questions

**Comments to the Author**

1. Is the manuscript technically sound, and do the data support the conclusions?

Reviewer #1: Yes

Reviewer #2: Yes

2. Has the statistical analysis been performed appropriately and rigorously? 

Reviewer #1: Yes

Reviewer #2: Yes

3. Have the authors made all data underlying the findings in their manuscript fully available?

Reviewer #1: Yes

Reviewer #2: Yes

4. Is the manuscript presented in an intelligible fashion and written in standard English?

Reviewer #1: Yes

Reviewer #2: Yes

5. Review Comments to the Author

Reviewer #1: Overall well written manuscript, addresses a clinically important issue for which a significant amount of controversy exists.

Only a few suggestions:

1. Clarify in the result section and Table #1, when describing most common causes of brain death "Traumatic hemorrhage" and "Non traumatic hemorrhage". Please clarify that these are intracranial hemorrhages or describe appropriately

2. Unclear if within 5 minutes all patients had a PaCO2 above 60 mmHg. If that is the case, state that all patients achieve this goal. If not, stare the percent that did.

Reviewer #2: A standard time interval for the apnea test has been suggested by national societies but not formalized by the international medical community. This paper is an attempt at starting that process, but because of varying ethical and legal standards, one paper will not complete the process. The scientific and statistical methods used by the authors are sound and are supported by the provided data. The choice of the smallest time interval that is satisfactory from a PaCO2 endpoint for protection of organs for transplantation is understandable but may be hard to achieve consensus. The additional time suggested by the American Academy of Neurology should not incur too great an oxygen debt or acidemia. As shown by this paper, the hemodynamics are usually easily controlled with vasoactive infusions. Some patients will in fact have intact brainstem reflexes that were not appreciated until the apnea test. Before the patient is declared dead, physicians still owe them a fiduciary duty. As suggested in this paper, future work should also focus on standardizing the balance of the protocol of the apnea test. The presented protocol is reasonable if the additional time is added.

6. PLOS authors have the option to publish the peer review history of their article (what does this mean?). If published, this will include your full peer review and any attached files.

Reviewer #1: No

Reviewer #2: **Yes: **J. Sean Funston, M.D.

---

## [Author Response · Author response to Decision Letter 0]

6 Jul 2022

Responses to reviewers’ comments

The authors would like to thank the reviewers for their constructive critique to improve the manuscript. We have made every effort to address the issues raised and to respond to all comments. The revisions are indicated in red font in the revised manuscript. Please, find next a detailed, point-by-point response to the reviewers’ comments. We hope that our revisions would meet the reviewers’ expectations.

Sincerely, 

Kyung Sook Hong

E-mail: hongks@ewha.ac.kr

Reviewer #1

1. Clarify in the result section and Table #1, when describing most common causes of brain death "Traumatic hemorrhage" and "Non traumatic hemorrhage". Please clarify that these are intracranial hemorrhages or describe

Response: We would like to thank the reviewer for evaluating our manuscript and for the insightful comment. Please note that both “traumatic hemorrhage” and “non-traumatic hemorrhage” are intracranial hemorrhages. As per the reviewer’s suggestion, we have added the relevant information to Table 1.

2. Unclear if within 5 minutes all patients had a PaCO2 above 60 mmHg. If that is the case, state that all patients achieve this goal. If not, stare the percent that did.

Response: We would like to thank the reviewer for the kind comments to improve our manuscript. To convey accurate information to our readers, we have added relevant contents, as per the reviewer’s comments. Additionally, the positive criterion for apnea test in Korea is a PaCO2 level of ≥50 mmHg. Therefore, PaCO2 levels >60 mmHg were not observed in all patients. As per the reviewer’s suggestion, we have added the following sentence to the revised manuscript: 

“In this study, the PaCO2 level exceeded 60 mmHg within 5 min in 76 (88.4%) and 81 (94.2%) cases of the 1st and 2nd apnea tests, respectively.”

Reviewer #2

1. A standard time interval for the apnea test has been suggested by national societies but not formalized by the international medical community. This paper is an attempt at starting that process, but because of varying ethical and legal standards, one paper will not complete the process. The scientific and statistical methods used by the authors are sound and are supported by the provided data. The choice of the smallest time interval that is satisfactory from a PaCO2 endpoint for protection of organs for transplantation is understandable but may be hard to achieve consensus. The additional time suggested by the American Academy of Neurology should not incur too great an oxygen debt or acidemia. As shown by this paper, the hemodynamics are usually easily controlled with vasoactive infusions. Some patients will in fact have intact brainstem reflexes that were not appreciated until the apnea test. Before the patient is declared dead, physicians still owe them a fiduciary duty. As suggested in this paper, future work should also focus on standardizing the balance of the protocol of the apnea test. The presented protocol is reasonable if the additional time is added.

Response: We would like to thank the reviewer for evaluating our manuscript and for his/her insightful comment. In the conclusion, we have stated that the appropriate duration of the apnea test is approximately 5 min. There are various debates concerning the duration of the apnea test for determination of brain death in Korea. As per the reviewer’s comments, we agree that the determination of brain death must be accompanied by ethical considerations. Therefore, as a further study, we will investigate the status of apnea tests in various institutions of Korea and plan ethical and reasonable suggestions for the positive criteria for apnea tests, including duration.

---

## [Editor Report · Decision Letter 1]

13 Jul 2022

Optimal Duration of the Apnea Test for Determining Brain Death: Benefit of the Short-Term Apnea Test

PONE-D-22-01880R1

Dear Dr. Hong,

We’re pleased to inform you that your manuscript has been judged scientifically suitable for publication and will be formally accepted for publication once it meets all outstanding technical requirements.

Kind regards,

Steven Eric Wolf, MD

Academic Editor

PLOS ONE
---

## [Editor Report · Acceptance letter]

19 Jul 2022

PONE-D-22-01880R1 

Optimal Duration of the Apnea Test for Determining Brain Death: Benefit of the Short-Term Apnea Test 

Dear Dr. Hong:

I'm pleased to inform you that your manuscript has been deemed suitable for publication in PLOS ONE. Congratulations! Your manuscript is now with our production department. 

Kind regards, 

on behalf of

Dr. Steven Eric Wolf 

Academic Editor

PLOS ONE